# Interpretable and Globally Optimal Prediction for Textual Grounding using Image Concepts

**Raymond A. Yeh,   Jinjun Xiong**[†]**,   Wen-mei W. Hwu,**

**Minh N. Do,   Alexander G. Schwing**

Department of Electrical Engineering, University of Illinois at Urbana-Champaign
[†]IBM Thomas J. Watson Research Center

`yeh17@illinois.edu, jinjun@us.ibm.com, w-hwu@illinois.edu,`
`minhdo@illinois.edu, aschwing@illinois.edu`

## Abstract

Textual grounding is an important but challenging task for human-computer inter­action, robotics and knowledge mining. Existing algorithms generally formulate the task as selection from a set of bounding box proposals obtained from deep net based systems. In this work, we demonstrate that we can cast the problem of textual grounding into a unified framework that permits efficient search over all possible bounding boxes. Hence, the method is able to consider significantly more proposals and doesn't rely on a successful first stage hypothesizing bounding box proposals. Beyond, we demonstrate that the trained parameters of our model can be used as word-embeddings which capture spatial-image relationships and provide interpretability. Lastly, at the time of submission, our approach outperformed the current state-of-the-art methods on the Flickr 30k Entities and the ReferItGame dataset by 3.08% and 7.77% respectively.

## 1   Introduction

Grounding of textual phrases, *i.e.*, finding bounding boxes in images which relate to textual phrases, is an important problem for human-computer interaction, robotics and mining of knowledge bases, three applications that are of increasing importance when considering autonomous systems, augmented and virtual reality environments. For example, we may want to guide an autonomous system by using phrases such as 'the bottle on your left,' or 'the plate in the top shelf.' While those phrases are easy to interpret for a human, they pose significant challenges for present day textual grounding algorithms, as interpretation of those phrases requires an understanding of objects and their relations.

Existing approaches for textual grounding, such as [38, 15] take advantage of the cognitive per­formance improvements obtained from deep net features. More specifically, deep net models are designed to extract features from given bounding boxes and textual data, which are then compared to measure their fitness. To obtain suitable bounding boxes, many of the textual grounding frameworks, such as [38, 15], make use of region proposals. While being easy to obtain, automatic extraction of region proposals is limiting, because the performance of the visual grounding is inherently constrained by the quality of the proposal generation procedure.

In this work we describe an interpretable mechanism which additionally alleviates any issues arising due to a limited number of region proposals. Our approach is based on a number of 'image concepts' such as semantic segmentations, detections and priors for any number of objects of interest. Based on those 'image concepts' which are represented as score maps, we formulate textual grounding as a search over all possible bounding boxes. We find the bounding box with highest accumulated score contained in its interior. The search for this box can be solved via an efficient branch and bound

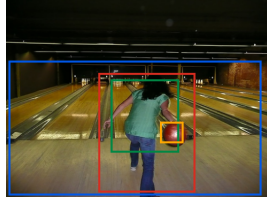 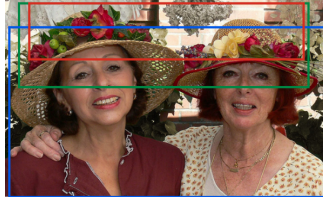 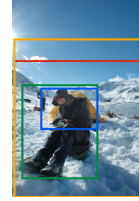

A woman in a green shirt is getting ready to throw her bowling ball down the lane...

Two women wearing hats covered in flowers are posing.

Young man wearing a hooded jacket sitting on snow in front of mountain area.

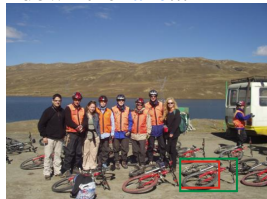 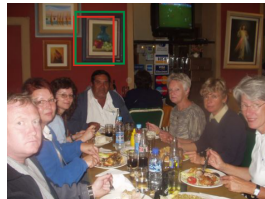 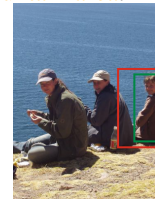

second bike from right in front

painting next to the two on the left

person all the way to the right

Figure 1: Results on the test set for grounding of textual phrases using our branch and bound based algorithm. **Top Row:** Flickr 30k Entities Dataset. **Bottom Row:** ReferItGame Dataset (Groundtruth box in green and predicted box in red).

scheme akin to the seminal efficient subwindow search of Lampert *et al*. [25]. The learned weights can additionally be used as word embeddings. We are not aware of any method that solves textual grounding in a manner similar to our approach and hope to inspire future research into the direction of deep nets combined with powerful inference algorithms.

We evaluate our proposed approach on the challenging ReferItGame [20] and the Flickr 30k Entities dataset [35], obtaining results like the ones visualized in Fig. 1. At the time of submission, our approach outperformed state-of-the-art techniques on the ReferItGame and Flickr 30k Entities dataset by 7.77% and 3.08% respectively using the IoU metric. We also demonstrate that the trained parameters of our model can be used as a word-embedding which captures spatial-image relationships and provides interpretability.

## 2   Related Work

**Textual grounding:** Related to textual grounding is work on image retrieval. Classical approaches learn a ranking function using recurrent neural nets [30, 6], or metric learning [13], correlation analysis [22], and neural net embeddings [9, 21]. Beyond work in image retrieval, a variety of techniques have been considered to explicitly ground natural language in images and video. One of the first models in this area was presented in [31, 24]. The authors describe an approach that jointly learns visual classifiers and semantic parsers.

Gong *et al*. [10] propose a canonical correlation analysis technique to associate images with descriptive sentences using a latent embedding space. In spirit similar is work by Wang *et al*. [42], which learns a structure-preserving embedding for image-sentence retrieval. It can be applied to phrase localization using a ranking framework. In [11], text is generated for a set of candidate object regions which is subsequently compared to a query. The reverse operation, *i.e.*, generating visual features from query text which is subsequently matched to image regions is discussed in [1].

In [23], 3D cuboids are aligned to a set of 21 nouns relevant to indoor scenes using a Markov random field based technique. A method for grounding of scene graph queries in images is presented in [17]. Grounding of dependency tree relations is discussed in [19] and reformulated using recurrent nets in [18]. Subject-Verb-Object phrases are considered in [39] to develop a visual knowledge extraction system. Their algorithm reasons about the spatial consistency of the configurations of the involved entities. In [15, 29] caption generation techniques are used to score a set of proposal boxes and returning the highest ranking one. To avoid application of a text generation pipeline on bounding box proposals, [38] improve the phrase encoding using a long short-term memory (LSTM) [12] based deep net. Additional modeling of object context relationship were explored in [32, 14]. Video

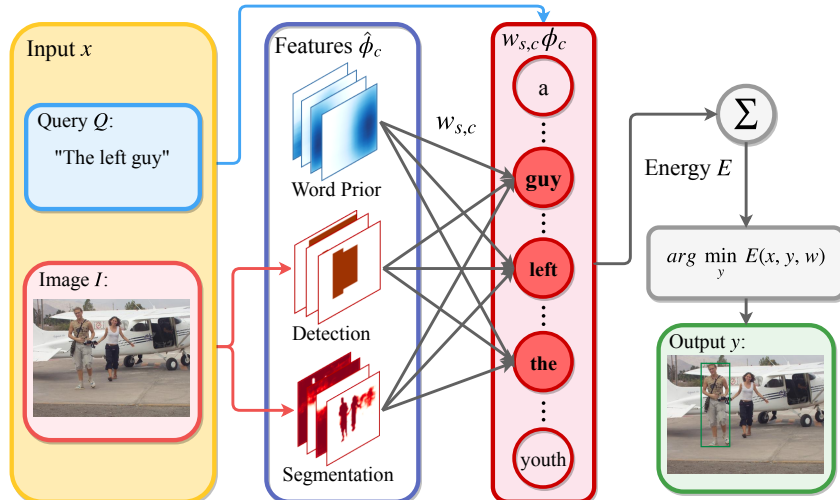

Figure 2: Overview of our proposed approach: We obtain word priors from the input query, take into account geometric features, as well as semantic segmentation features computed from the provided input image. We compute the three image cues to predict the four variables of the bounding box $y = (y_1, \ldots, y_4)$.

datasets, although not directly related to our work in this paper, were used for spatiotemporal language grounding in [27, 45].

Common datasets for visual grounding are the ReferItGame dataset [20] and a newly introduced Flickr 30k Entities dataset [35], which provides bounding box annotations for noun phrases of the original Flickr 30k dataset [44].

In contrast to all of the aforementioned methods, which are largely based on region proposals, we suggest usage of efficient subwindow search as a suitable inference engine.

**Efficient subwindow search:** Efficient subwindow search was proposed by Lampert *et al.* [25] for object localization. It is based on an extremely effective branch and bound scheme that can be applied to a large class of energy functions. The approach has been applied to very efficient deformable part models [43], for object class detection [26], for weakly supervised localization [5], indoor scene understanding [40], diverse object proposals [41] and also for spatio-temporal object detection proposals [33].

## 3 Exact Inference for Grounding

We outline our approach for textual grounding in Fig. 2. In contrast to the aforementioned techniques for textual grounding, which typically use a small set of bounding box proposals, we formulate our language grounding approach as an energy minimization over a large number of bounding boxes. The search over a large number of bounding boxes allows us to retrieve an accurate bounding-box prediction for a given phrase and an image. Importantly, by leveraging efficient branch-and-bound techniques, we are able to find the global minimizer for a given energy function very effectively.

Our energy is based on a set of 'image concepts' like semantic segmentations, detections or image priors. All those concepts come in the form of score maps which we combine linearly before searching for the bounding box containing the highest accumulated score over the combined score map. It is trivial to add additional information to our approach by adding additional score maps. Moreover, linear combination of score maps reveals importance of score maps for specific queries as well as similarity between queries such as 'skier' and 'snowboarder.' Hence the framework that we discuss in the following is easy to interpret and extend to other settings.

**General problem formulation:** For simplicity we use $x$ to refer to both given input data modalities, *i.e.*, $x = (Q, I)$, with query text, $Q$, and image, $I$. We will differentiate them in the narrative. In addition, we define a bounding box $y$ via its top left corner $(y_1, y_2)$ and its bottom right corner $(y_3, y_4)$ and subsume the four variables of interest in the tuple $y = (y_1, \ldots, y_4) \in \mathcal{Y} = \prod_{i=1}^{4}\{0, \ldots, y_{i,\max}\}$. Every integral coordinate $y_i$, $i \in \{1, \ldots, 4\}$ lies within the set $\{0, \ldots, y_{i,\max}\}$, and $\mathcal{Y}$ denotes the

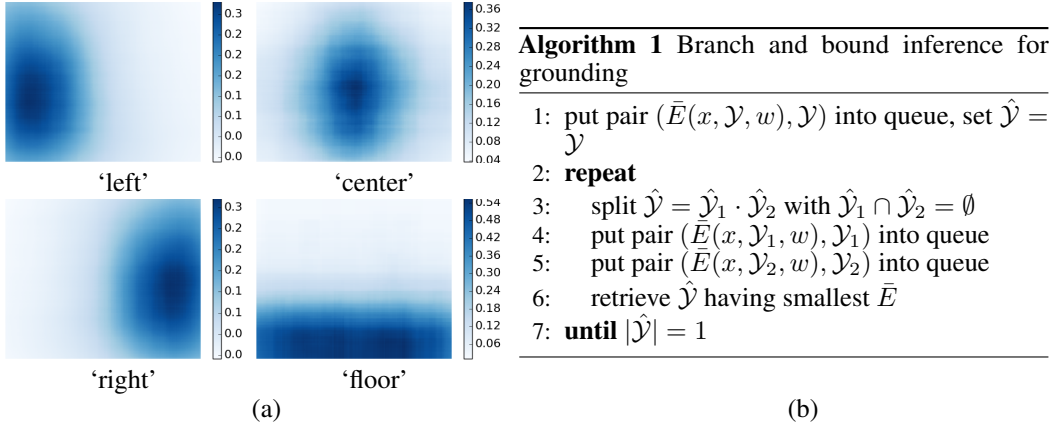

Figure 3: Word priors in (a) and the employed inference algorithm in (b).

product space of all four coordinates. For notational simplicity only, we assume all images to be scaled to identical dimensions, *i.e.*, $y_{i,\max}$ is not dependent on the input data $x$. We obtain a bounding box prediction $\hat{y}$ given our data $x$, by solving the energy minimization

$$\hat{y} = \arg\min_{y \in \mathcal{Y}} E(x, y, w), \tag{1}$$

to global optimality. Note that $w$ refers to the parameters of our model. Despite the fact that we are 'only' interested in a single bounding box, the product space $\mathcal{Y}$ is generally too large for exhaustive minimization of the energy specified in Eq. (1). Therefore, we pursue a branch-and-bound technique in the following.

To apply branch and bound, we assume that the energy function $E(x, y, w)$ depends on two sets of parameters $w = [w_t^T, w_r^T]^T$, *i.e.*, the top layer parameters $w_t$ of a neural net, and the remaining parameters $w_r$. In light of this decomposition, our approach requires the energy function to be of the following form:

$$E(x, y, w) = w_t^T \phi(x, y, w_r).$$

Note that the features $\phi(x, y, w_r)$ may still depend non-linearly on all but the top-layer parameters. This assumption does not pose a severe restriction since almost all of the present-day deep net models typically obtain the logits $E(x, y, w)$ using a fully-connected layer or a convolutional layer with kernel size $1 \times 1$ as the last computation.

**Energy Function Details:** Our energy function $E(x, y, w)$ is based on a set of 'image concepts,' such as semantic segmentation of object categories, detections, or word priors, all of which we subsume in the set $\mathcal{C}$. Importantly, all image concepts $c \in \mathcal{C}$ are attached a parametric score map $\hat{\phi}_c(x, w_r) \in \mathbb{R}^{W \times H}$ following the image width $W$ and height $H$. Note that those parametric score maps may depend nonlinearly on some parameters $w_r$. Given a bounding box $y$, we use the scalar $\phi_c(x, y, w_r) \in \mathbb{R}$ to refer to the score accumulated within the bounding box $y$ of score map $\hat{\phi}_c(x, w_r)$.

To define the energy function we also introduce a set of words of interest, *i.e.*, $\mathcal{S}$. Note that this set contains a special symbol denoting all other words not of interest for the considered task. We use the given query $Q$, which is part of the data $x$, to construct indicators, $\iota_s = \delta(s \in Q) \in \{0, 1\}$, denoting for every token $s \in \mathcal{S}$ its existence in the query $Q$, where $\delta$ denotes the indicator function.

Based on this definition, we formulate the energy function as follows:

$$E(x, y, w) = \sum_{s \in \mathcal{S} : \iota_s = 1} \sum_{c \in \mathcal{C}} w_{s,c} \phi_c(x, y, w_r), \tag{2}$$

where $w_{s,c}$ is a parameter connecting a word $s \in \mathcal{S}$ to an image concept $c \in \mathcal{C}$. In other words, $w_t = (w_{s,c} : \forall s \in \mathcal{S}, c \in \mathcal{C})$. This energy function results in a sparse $w_t$, which increases the speed of inference.

**Score maps:** The energy is given by a linear combination of accumulated score maps $\phi_c(x, y, w_r)$. In our case, we use $|\mathcal{C}| = k_1 + k_2 + k_3$ of those maps, which capture three kinds of information: (i) $k_1$ word-priors; (ii) $k_2$ geometric information cues; and (iii) $k_3$ image based segmentations and detections. We discuss each of those maps in the following.

| Approach | Accuracy (%) |
|---|---|
| SCRC (2016) [15] | 27.80 |
| DSPE (2016) [42] | 43.89 |
| GroundeR (2016) [38] | 47.81 |
| CCA (2017) [36] | 50.89 |
| Ours (Prior + Geo + Seg + Det) | **51.63** |
| Ours (Prior + Geo + Seg + bDet) | **53.97** |

Table 1: Phrase localization performance on Flickr 30k Entities.

| Approach | Accuracy (%) |
|---|---|
| SCRC (2016) [15] | 17.93 |
| GroundeR (2016) [38] | 23.44 |
| GroundeR (2016) [38] +SPAT | 26.93 |
| Ours (Prior + Geo) | 25.56 |
| Ours (Prior + Geo + Seg) | **33.36** |
| Ours (Prior + Geo + Seg + Det) | **34.70** |

Table 2: Phrase localization performance on ReferItGame.

| | people | clothing | body parts | animals | vehicles | instruments | scene | other |
|---|---|---|---|---|---|---|---|---|
| # Instances | 5,656 | 2,306 | 523 | 518 | 400 | 162 | 1,619 | 3,374 |
| GroundeR(2016) [38] | 61.00 | 38.12 | 10.33 | 62.55 | 68.75 | 36.42 | 58.18 | 29.08 |
| CCA(2017) [36] | 64.73 | **46.88** | 17.21 | 65.83 | 68.75 | 37.65 | 51.39 | 31.77 |
| Ours | **68.71** | 46.83 | **19.50** | **70.07** | **73.75** | **39.50** | **60.38** | **32.45** |

Table 3: Phrase localization performance over types on Flickr 30k Entities (accuracy in %).

For the top $k_1$ words in the training set we construct word prior maps like the ones shown in Fig. 3 (a). To obtain the prior for a particular word, we search a given training set for each occurrence of the word. With the corresponding subset of image-text pairs and respective bounding box annotations at hand, we compute the average number of times a pixel is covered by a bounding box. To facilitate this operation, we scale each image to a predetermined size. Investigating the obtained word priors given in Fig. 3 (a) more carefully, it is immediately apparent that they provide accurate location information for many of the words.

The $k_2 = 2$ geometric cues provide the aspect ratio and the area of the hypothesized bounding box $y$. Note that the word priors and geometry features contain no information about the image specifics.

To encode measurements dedicated to the image at hand, we take advantage of semantic segmentation and object detection techniques. The $k_3$ image based features are computed using deep neural nets as proposed by [4, 37, 2]. We obtain probability maps for a set of class categories, *i.e.*, a subset of the nouns of interest. The feature $\phi$ accumulates the scores within the hypothesized bounding box $y$.

**Inference:** The algorithm to find the bounding box $\hat{y}$ with lowest energy as specified in Eq. (1) is based on an iterative decomposition of the output space $\mathcal{Y}$ [25], summarized in Fig. 3 (b). To this end we search across subsets of the product space $\mathcal{Y}$ and we define for every coordinate $y_i$, $i \in \{1, \ldots, 4\}$ a corresponding lower and upper bound, $y_{i,\text{low}}$ and $y_{i,\text{high}}$ respectively. More specifically, considering the initial set of all possible bounding boxes $\mathcal{Y}$, we divide it into two disjoint subsets $\hat{\mathcal{Y}}_1$ and $\hat{\mathcal{Y}}_2$. For example, by constraining $y_1$ to $\{0, \ldots, y_{1,\text{max}}/2\}$ and $\{y_{1,\text{max}}/2 + 1, \ldots, y_{1,\text{max}}\}$ for $\hat{\mathcal{Y}}_1$ and $\hat{\mathcal{Y}}_2$ respectively, while keeping all the other intervals unchanged. It is easy to see that we can repeat this decomposition by choosing the largest among the four intervals and recursively dividing it into two parts.

Given such a repetitive decomposition strategy for the output space, and since the energy $E(x, y, w)$ for a bounding box $y$ is obtained using a linear combination of word priors and accumulated segmentation masks, we can design an efficient branch and bound based search algorithm to exactly solve the inference problem specified in Eq. (1). The algorithm proceeds by iteratively decomposing a product space $\hat{\mathcal{Y}}$ into two subspaces $\hat{\mathcal{Y}}_1$ and $\hat{\mathcal{Y}}_2$. For each subspace, the algorithm computes a lower bound $\bar{E}(x, \mathcal{Y}_j, w)$ for the energy of all possible bounding boxes within the respective subspace. Intuitively, we then know, that any bounding box within the subspace $\hat{\mathcal{Y}}_j$ has a larger energy than the lower bound. The algorithm proceeds by choosing the subspace with lowest lower-bound until this subspace consists of a single element, *i.e.*, until $|\hat{\mathcal{Y}}| = 1$. We summarize this algorithm in Alg. 1 (Fig. 3 (b)).

To this end, it remains to show how to compute a lower bound $\bar{E}(x, \mathcal{Y}_j, w)$ on the energy for an output space, and to illustrate the conditions which guarantee convergence to the global minimum of the energy function.

For the latter, we note that two conditions are required to ensure convergence to the optimum: (i) the bound of the considered product space has to lower-bound the true energy for each of its bounding

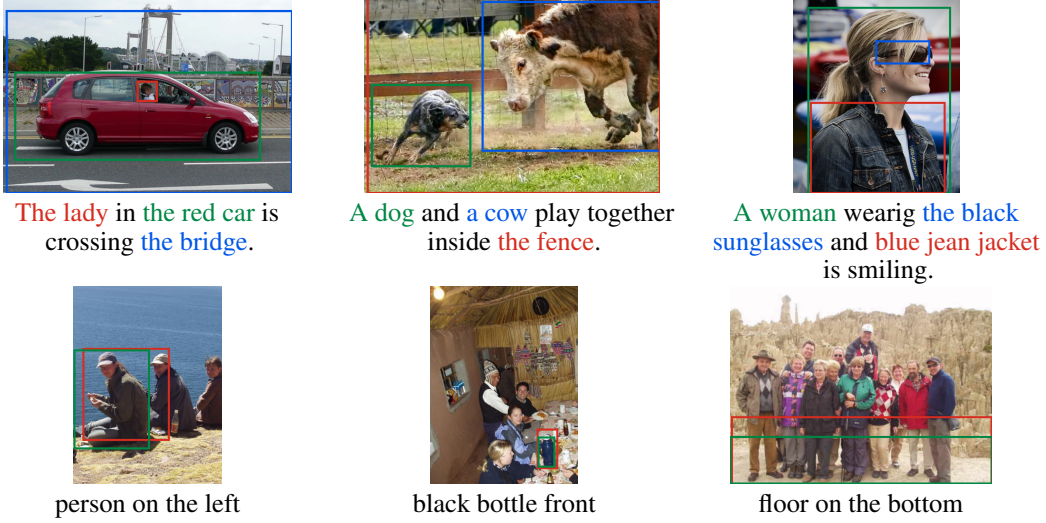

The lady in the red car is crossing the bridge.     A dog and a cow play together inside the fence.     A woman wearig the black sunglasses and blue jean jacket is smiling.

person on the left     black bottle front     floor on the bottom

Figure 4: Results on the test set for grounding of textual phrases using our branch and bound based algorithm. **Top Row:** Flickr 30k Entities Dataset. **Bottom Row:** ReferItGame Dataset (Groundtruth box in green and predicted box in red).

box hypothesis $\hat{y} \in \hat{\mathcal{Y}}$, *i.e.*, $\forall \hat{y} \in \hat{\mathcal{Y}}$, $\bar{E}(x, \hat{\mathcal{Y}}, w) \leq E(x, \hat{y}, w)$; (ii) the bound has to be exact for all possible bounding boxes $y \in \mathcal{Y}$, *i.e.*, $\bar{E}(x, y, w) = E(x, y, w)$. Given those two conditions, global convergence of the algorithm summarized in Alg. 1 is apparent: upon termination we obtain an 'interval' containing a single bounding box, and its energy is at least as low as the one for any other interval.

For the former, we note that bounds on score maps for bounding box intervals can be computed by considering either the largest or the smallest possible bounding box in the bounding box hypothesis, $\hat{\mathcal{Y}}$, depending on whether the corresponding weight in $w_t$ is positive or negative and whether the feature maps contain only positive or negative values. Intuitively, if the weight is positive and the feature mask contains only positive values, we obtain the smallest lower bound $\bar{E}(x, \hat{\mathcal{Y}}, w)$ by considering the content within the smallest possible bounding box. Note that the score maps do not necessarily contain only positive or negative numbers. However we can split the given score maps into two separate score maps (*i.e.*, one with only positive values, and another with only negative values) while applying the same weight.

It is important to note that computation of the bound $\bar{E}(x, \hat{\mathcal{Y}}, w)$ has to be extremely effective for the algorithm to run at a reasonable speed. However, computing the feature mask content for a bounding box is trivially possible using integral images. This results in a constant time evaluation of the bound, which is a necessity for the success of the branch and bound procedure.

**Learning the Parameters:** With the branch and bound based inference procedure at hand, we now describe how to formulate the learning task. Support-vector machine intuition can be applied. Formally, we are given a training set $\mathcal{D} = \{(x, y)\}$ containing pairs of input data $x$ and groundtruth bounding boxes $y$. We want to find the parameters $w$ of the energy function $E(x, y, w)$ such that the energy of the groundtruth is smaller than the energy of any other configuration. Negating this statement results in the following desiderata when including an additional margin term $L(y, \hat{y})$, also known as task-loss, which measures the loss between the groundtruth $y$ and another configuration $\hat{y}$:

$$-E(x, y, w) \geq -E(x, \hat{y}, w) + L(\hat{y}, y) \quad \forall \hat{y} \in \mathcal{Y}.$$

Since we want to enforce this inequality for all configurations $\hat{y} \in \mathcal{Y}$, we can reduce the number of constraints by enforcing it for the highest scoring right hand side. We then design a cost function which penalizes violation of this requirement linearly. We obtain the following structured support vector machine based surrogate loss minimization:

$$\min_{w} \quad \frac{C}{2} \|w\|_2^2 + \sum_{(x,y) \in \mathcal{D}} \max_{\hat{y} \in \mathcal{Y}} \left( -E(x, \hat{y}, w) + L(\hat{y}, y) \right) + E(x, y, w) \tag{3}$$

where $C$ is a hyperparameter adjusting the squared norm regularization to the data term. For the task loss $L(\hat{y}, y)$ we use intersection over union (IoU).

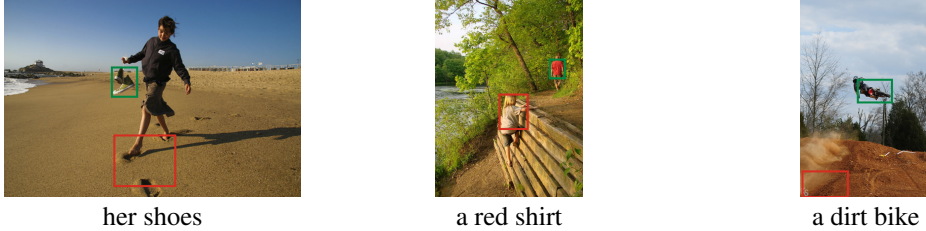

| her shoes | a red shirt | a dirt bike |

Figure 5: Flickr 30k Failure Cases. (Green box: ground-truth, Red box:predicted)

By fixing the parameters $w_r$ and only learning the top layer parameters $w_t$, Eq. (3) is equivalent to the problem of training a structured SVM. We found the cutting-plane algorithm [16] to work well in our context. The cutting-plane algorithm involves solving the maximization task. This maximization over the output space $\mathcal{Y}$ is commonly referred to as loss-augmented inference. Loss augmented inference is structurally similar to the inference task given in Eq. (1). Since maximization is identical to negated minimization, the computation of the bounds for the energy $E(x, \hat{y}, w)$ remains identical. To bound the IoU loss, we note that a quotient can be bounded by bounding nominator and denominator independently. To lower bound the intersection of the groundtruth box with the hypothesis space we use the smallest hypothesized bounding box. To upper bound the union of the groundtruth box with the hypothesis space we use the largest bounding box.

Further, even though not employed to obtain the results in this paper, we mention that it is possible to backpropagate through the neural net parameters $w_r$ that influence the energy non-linearly. This underlines that our initial assumption is merely a construct to design an effective inference procedure.

## 4   Experimental Evaluation

In the following we first provide additional details of our implementation before discussing the results of our approach.

**Language processing:** In order to process free-form textual phrases efficiently, we restricted the vocabulary size to the top 200 most frequent words in the training set for the ReferItGame, and to the top 1000 most frequent training set words for Flickr 30k Entities; both choices cover about 90% of all phrases in the training set. We map all the remaining words into an additional token. We don't differentiate between uppercase and lower case characters and we also ignore punctuation.

**Segmentation and detection maps:** We employ semantic segmentation, object detection, and pose-estimation. For segmentation, we use the DeepLab system [4], trained on PASCAL VOC-2012 [8] semantic image segmentation task, to extract the probability maps for 21 categories. For detection, we use the YOLO object detection system [37], to extract 101 categories, 21 trained on PASCAL VOC-2012, and 80 trained on MSCOCO [28]. For pose estimation, we use the system from [2] to extract the body part location, then post-process to get the head, upper body, lower body, and hand regions.

For the ReferItGame, we further fine-tuned the last layer of the DeepLab system to include the categories of 'sky,' 'ground,' 'building,' 'water,' 'tree,' and 'grass.' For the Flickr 30k Entities, we also fine-tuned the last layer of the DeepLab system using the eight coarse-grained types and eleven colors from [36].

**Preprocessing and post-processing:** For word prior feature maps and the semantic segmentation maps, we take an element-wise logarithm to convert the normalized feature counts into log-probabilities. The summation over a bounding box region then retains the notion of a joint log-probability. We also centered the feature maps to be zero-mean, which corresponds to choosing an initial decision threshold. The feature maps are resized to dimension of $64 \times 64$ for efficient computation, and the predicted box is scaled back to the original image dimension during evaluation. We re-center the prediction box by a constant amount determined using the validation set, as resizing truncate box coordinates to an integer.

**Efficient sub-window search implementation:** In order for the efficient subwindow search to run at a reasonable speed, the lower bound on $E$ needs to be computed as fast as possible. Observe that, $E(x, y, w)$, is a weighted sum of the feature maps over the region specified by a hypothesized bounding box. To make this computation efficient, we pre-compute integral images. Given an integral

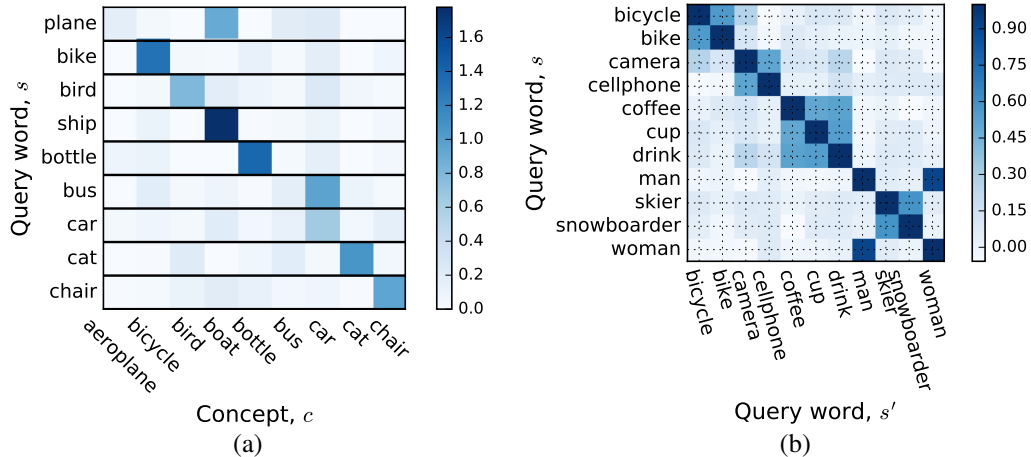

Figure 6: (a) Trained weight, $w_{s,c}$, visualization on words, $s$ and segmentation concepts, $c$, on Flicker 30k. (b) Cosine similairty visualization between words vector, $w_s$ and $w'_s$ on Flicker 30k.

image, the computation for each of the bounding box is simply a look-up operation. This trick can similarly be applied for the geometric features. Since we know the range of the ratio and areas of the bounding boxes ahead of time, we cache the results in a look up table as well.

The **ReferItGame dataset** consists of more than 99,000 regions from 20,000 images. Bounding boxes are assigned to natural language expressions. We use the same bounding boxes as [38] and the same training test set split, *i.e.*, 10,000 images for testing, 9,000 images for training and 1,000 images for validation.

The **Flickr 30k Entities dataset** consists of more than 275k bounding boxes from 31k image, where each bounding box is annotated with the corresponding natural language phrase. We us the same training, validation and testing split as in [35].

**Quantitative evaluation:** In Tab. 1 and Tab. 2 we quantitatively compare the results of our approach to recent state-of-the-art baselines, where Prior = word priors, Geo = geometric information, Seg = Segmentation maps, Det = Detection maps, bDet = Detection maps + body parts detection. An example is considered as correct, if the predicted box overlaps with the ground-truth box by more than 0.5 IoU. We observe our approach to outperform competing methods by around 3% on the Flickr 30k Entities dataset and by around 7% on the ReferItGame dataset.

We also provide an ablation study of the word and image information as shown in Tab. 1 and Tab. 2.

In Tab. 3 we analyze the results for each "phrase type" provided by Flicker30k Entities dataset. As can be seen, our system outperforms the state-of-the-art in all phrase types except for clothing.

We note that our results have been surpassed by [3, 7, 34], where they fine-tuned the entire network including the feature extractions or trained more feature detectors; CCA, GroundeR and our approach uses a fixed pre-trained network for extracting image features.

**Qualitative evaluation:** Next we evaluate our approach qualitatively. In Fig. 1 and Fig. 4 we show success cases. We observe that our method successfully captures a variety of objects and scenes. In Fig. 5 we illustrate failure cases. We observe that for a few cases word prior may hurt the prediction (*e.g.*, shoes are typically on the bottom half of the image.) Also our system may fail when the energy is not a linear combination of the feature scores. For example, the score of "dirt bike" should not be the score of "dirt" + the score of "bike." We provide additional results in the supplementary material.

**Learned parameters + word embedding:** Recall, in Eq. (2), our model learns a parameter per phrase word and concept pair, $w_{s,c}$. We visualize its magnitude in Fig. 6 (a) for a subset of words and concepts. As can be seen, $w_{s,c}$ is large, when the phrase word and the concept are related, (*e.g. s =* ship and $c$ = boat). This demonstrates that our model successfully learns the relationship between phrase words and image concepts. This also means that the "word vector," $w_s = [w_{s,1}, w_{s,2}, ...w_{s,|\mathcal{C}|}]$, can be interpreted as a word embedding. Therefore, in Fig. 6 (b), we visualize the cosine similarity between pairs of word vectors. Expected groups of words form, for example (bicycle, bike), (camera, cellphone), (coffee, cup, drink), (man woman), (snowboarder, skier). The word vectors capture

image-spatial relationship of the words, meaning items that can be "replaced" in an image are similar; (*e.g.*, a "snowboarder" can be replaced with a "skier" and the overall image would still be reasonable).

**Computational Efficiency:** Overall, our method's inference speed is comparable to CCA and much faster than GroundeR. The inference speed can be divided into three main parts, (1) extracting image features, (2) extracting language features, and (3) computing scores. For extracting image features, GroundeR requires a forward pass on VGG16 for each image region, where CCA and our approach requires a single forward pass which can be done in 142.85 ms. For extracting language features, our method requires index lookups, which takes negligible amount of time (less than 1e-6 ms). CCA, uses Word2vec for processing the text, which takes 0.070 ms. GroundeR uses a Long-Short-Term Memory net, which takes 0.7457 ms. Computing the scores with our C++ implementation takes 1.05ms on a CPU. CCA needs to compare projections of the text and image features, which takes 13.41ms on a GPU and 609ms on a CPU. GroundeR uses a single fully connected layer, which takes 0.31 ms on a GPU.

## 5 Conclusion

We demonstrated a mechanism for grounding of textual phrases which provides interpretability, is easy to extend, and permits globally optimal inference. In contrast to existing approaches which are generally based on a small set of bounding box proposals, we efficiently search over all possible bounding boxes. We think interpretability, *i.e.*, linking of word and image concepts, is an important concept, particularly for textual grounding, which deserves more attention.

**Acknowledgments:** This material is based upon work supported in part by the National Science Foundation under Grant No. 1718221. This work is supported by NVIDIA Corporation with the donation of a GPU. This work is supported in part by IBM-ILLINOIS Center for Cognitive Computing Systems Research (C3SR) - a research collaboration as part of the IBM Cognitive Horizons Network.

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
