[Reviews · NeurIPS 2017]

Reviewer 1



The paper proposes a unified framework to solve the textual grounding problem as an energy minimization problem, and uses an efficient branch and bound scheme to find the global optimum. This differs from approaches that evaluate on generated region proposals. The energy function used in this framework is a linear combination of feature functions as score maps, each of either the form of word priors, geometric information cues, or image based segmentations and detections. It's easy to add more feature functions of the same form to the framework. The top-level weights combining the features functions are formulated and learned in the form of structured SVMs. There are many parts of the proposed system, including the many feature functions as score maps being combined in the energy function, weight learning as a structured SVM problem, global inference using the efficient sub-window search, etc., but the paper seems to have given enough details and references to describe all parts of the system. One thing I would be interested to know, not clearly given in the paper, is how efficient the inference is, exactly, in terms of computation resource and time.

Reviewer 2



The authors propose a method for textual grounding, that is identifying the correct object bounding box described by words that are in general ambiguous but specific enough withing the given image. The main advantage of the method is that unlike other methods it does not rely on having access to a good set of proposal bounding boxes. The authors show how to setup an energy function which can be global minized using efficient subwindow search (a branch-and-bound-based method). The energy function is a linear combination of "image concepts" weighted by the words contained in the query. These weights specify how much a given word relates to a given concept. The image concepts are features encoding priors of the most frequent words, probability maps for selected nouns obtained from a convolutional neural network, and two geometric cues on the size of the bounding box. Finally, the authors formulate the parameter learning as a structured support vector machine. The paper builds upon the efficient subwindow search method applying it to textual grounding in an interesting non-trivial way. The paper is in general well written, reasonably easy to understand, the experiments well done and the results seem convincing. There are two things that might be limiting: 1. The image concepts are combined linearly in the energy function which brings the problems with some queries, such as "dirt bike" discussed by the authors. However, it also brings efficient evaluation of the bound. 2. The runtime of the method. The authors should add a comparison of runtimes (or at least the runtime of their method) to the tables comparing the accuracy. Comments: L6: we able -> we are able Fig.2 vs. Eq.1: please use either argmax or argmin consistently L100+1: "to depends" -> depends L102: sever -> severe L113 and Eq.2: the indicator \iota_s seems superfluous, removing it and using iteration over s \in Q would make reading simpler L135: accumulate -> accumulates L235: us -> use

Reviewer 3



Summary The approach proposes a simple method to find the globally optimal (under the formulation) box in an image which represents the grounding of the textual concept. Unlike previous approaches which adopt a two stage pipeline where region proposals are first extracted and then combined into grounded image regions (see [A] for example), this approach proposes a formulation which finds the globally optimal box for a concept. The approach assumes the presence of spatial heat maps depicting various concepts, and uses priors and geometry related information to construct an energy function, with learnable parameters for combining different cues. A restriction of the approach is that known concepts can only be combined linearly with each other (for instance the score map of “dirt bike” is “dirt” + “bike” with the learn weighting ofcourse), but this also allows for optimal inference for the given model class. More concretely the paper proposes an efficient technique based on [22] to use branch and bound for efficient sub-window search. Training is straightforward and clean through cutting-plane training of structured SVM. The paper also shows how to do efficient loss augmented inference during SVM training which makes the same branch and bound approach applicable to cutting-plane training as well. Finally, results are shown against competitive (near- state of the art approaches) on two datasets where the proposed approach is shown to outperform the state of the art. Strengths - Approach alleviates the need for a blackbox stage which generates region proposals. - The interpretation of the weights of the model and the concepts as word embeddings is a neat little tidbit. - The paper does a good job of commenting on cases where the approach fails, specifically pointing out some interesting examples such as “dirt bike” where the additive nature of the feature maps is a limitation. - The paper has some very interesting ideas such as the use of the classic integral images technique to do efficient inference using branch and bound, principled training of the model via. a clean application of Structural SVM training with the cutting plane algorithm etc. Weakness 1. Paper misses citing a few relevant recent related works [A], [B], which could also benefit from the proposed technique and use region proposals. 2. Another highly relevant work is [C] which does efficient search for object proposals in a similar manner to this approach building on top of the work of Lampert et.al.[22] 3. It is unclear what SPAT means in Table. 2. 4. How was Fig. 6 b) created? Was it by random sub-sampling of concepts? 5. It would be interesting to consider a baseline which just uses the feature maps (used in the work, say shown in Fig. 2) and the phrases and simply regresses to the target coordinates using an MLP. Is it clear that the proposed approach would outperform it? (*) 6. L130: It was unclear to me how the geometry constraints are exactly implemented in the algorithm, i.e. the exposition of how the term k2 is computed was uncler. It would be great to provide details. Clear explanation of this seems especially important since the performance of the system seems highly dependent on this term (as it is trivial to maximize the sum of scores of say detection heat maps by considering the entire image as the set). Preliminary Evaluation The paper has a neat idea which is implemented in a very clean manner, and is easy to read. Concerns important for the rebuttal are marked with (*) above. [A] Hu, Ronghang, Marcus Rohrbach, Jacob Andreas, Trevor Darrell, and Kate Saenko. 2016. “Modeling Relationships in Referential Expressions with Compositional Modular Networks.” arXiv [cs.CV]. arXiv. http://arxiv.org/abs/1611.09978. [B] Nagaraja, Varun K., Vlad I. Morariu, and Larry S. Davis. 2016. “Modeling Context Between Objects for Referring Expression Understanding.” arXiv [cs.CV]. arXiv. http://arxiv.org/abs/1608.00525. [C] Sun, Qing, and Dhruv Batra. 2015. “SubmodBoxes: Near-Optimal Search for a Set of Diverse Object Proposals.” In Advances in Neural Information Processing Systems 28, edited by C. Cortes, N. D. Lawrence, D. D. Lee, M. Sugiyama, and R. Garnett, 1378–86. Curran Associates, Inc.